# Limitations of hormonal and clinical markers in predicting GnRH agonist trigger success in polycystic ovary syndrome: A critical reappraisal

Fazilet Kubra Boynukalin [1,2]*, Zalihe Yarkıner[3], Meral Gultomruk[4], Ozkan Ozdamar[1], Peter Humaidan[5], Mustafa Bahceci[1], Gurkan Bozdag[1]

1 Infertility Department, Bahçeci Fulya IVF Center, Istanbul, Turkey, 2 Department of Obstetrics and Gynecology, Uskudar University, Istanbul, Turkey, 3 Department of Basic Sciences, Faculty of Arts and Sciences, Cyprus International University, Kyrenia, Cyprus, 4 Research and Development Department, Bahçeci Health Group, Istanbul, Turkey, 5 The Fertility Clinic, Skive Regional Hospital, Faculty of Health, Aarhus University, Skive, Denmark

* kboynukalin@bahceci.com

## Abstract

### Background

Polycystic ovary syndrome (PCOS) is associated with altered hypothalamic-pituitary-ovarian function, which may affect the success of GnRH agonist triggers used during IVF. Identifying reliable predictors of oocyte yield in these patients remains a clinical challenge.

### Purpose

To evaluate possible predictors of suboptimal oocyte retrieval per aspirated follicle when ovulation is triggered with GnRH agonist in PCOS patients.

### Methods

Between 30/04/2021 and 30/12/2022, a prospective cohort study was conducted among women with PCOS (n = 104) in which a GnRH agonist trigger was employed during a GnRH antagonist protocol. Hormonal and clinical parameters were tested for their ability to predict the oocyte per aspirated follicle (OPF) rate in in-vitro fertilization (IVF) treatment.

### Results

The mean age of the patients, mean number of aspirated follicles, number of collected oocyte cumulus complexes and OPF were 28.6 ± 3.9 years, 31.4 ± 10.2, 21.7 ± 8.9, and 70.2 ± 19.1%, respectively. When patients were stratified into three groups according to their OPF percentiles (Q1:0–25th percentile, Q2:26–75th percentile, Q3:76–100th percentile), body mass index (BMI) and antral follicle count (AFC)

**Data availability statement:** The dataset is publicly accessible at the following repository: https://doi.org/10.6084/m9.figshare.30227524.

**Funding:** The author(s) received no specific funding for this work.

**Competing interests:** The authors have declared that no competing interests exist.

were significantly higher in the Q1 group compared to the Q2 and Q3 groups. However, regression analysis revealed that only AFC was an independent predictor of the OPF rate (RR: −0.005, 95% CI: −0.008 to −0.002, p = 0.001), but not BMI or serum LH levels on the day of triggering. Notably, the predictive validity of AFC to recognize a low OPF rate was poor (AUC = 0.561).

## Conclusion

A high AFC was the only identifiable predictor of the OPF rate and a suboptimal response when a GnRH agonist trigger was used for final follicular maturation. However, since a low AUC for AFC suggests a poor performance, we conclude that this study was not able to find any reliable prediction markers for the OPF rate in PCOS patients triggered with GnRHa.

---

## Introduction

Until recently human chorionic gonadotropin (hCG) was the preferred trigger for final follicular maturation in ovarian stimulation protocols during in-vitro fertilization (IVF) cycles. Although gonadotropin-releasing hormone (GnRH) agonist was initially introduced in the early 1970s for final oocyte maturation [1], its application became prominent with the advent of GnRH antagonist protocols and the adoption of freeze-all strategies in in-vitro fertilization (IVF) treatments. Importantly, early studies indicated that triggering with a GnRH agonist during GnRH antagonist co-treatment effectively eliminated the risk of ovarian hyperstimulation syndrome [2], which spurred significant clinical interest in this approach. However, concerns about the reproductive outcomes after GnRH agonist trigger in fresh transfer cycles were initially raised by two randomized controlled trials, reporting low pregnancy rates and high early pregnancy loss rates due to an insufficient luteal phase when a standard luteal phase support was used [3,4]. As improved cryopreservation techniques became a crucial base for 'segmentation', in which ovarian stimulation and oocyte retrieval are separated from the embryo transfer [5], the use of GnRH agonist trigger has evolved into a standard procedure, becoming a cornerstone of IVF practice by enhancing the safety of the treatment [6].

Although the GnRH agonist trigger protocol is preferred in segmented IVF cycles, some clinicians might still have concerns about the risk of a suboptimal response to GnRHa trigger, resulting in a suboptimal oocyte yield. However, the available data on the risk of a suboptimal response to GnRH agonist trigger is inconclusive due to heterogeneity in the definition [7–10]. Thus, in previous studies, a suboptimal response was defined as either i) a post-trigger LH value <15 mIU/ml, ii) decreased total number of oocytes, or iii) decreased oocyte yield as calculated by the ratio of the total number of oocytes retrieved to the total number of follicles >10 mm on the day of triggering. Although the first definition describes a laboratory condition and, thus, lack of subjectivity, the clinical results might not correlate very well. Nevertheless, the second and third definitions could also stem from technical challenges encountered

during the oocyte retrieval process, variations in the approach to puncturing each follicle, or selectivity based on follicular diameter. In addition, as quoted previously, there might be a trend for greater measurement errors in larger follicles [11] and a steady decrease in reliability of follicular measurements as the number of follicles increases [12]. In this context, oocyte per aspirated follicle (OPF) rather than follicles of any size on the day of triggering, might be a better approach to eliminate inherent biases.

Polycystic ovary syndrome (PCOS) is characterized by alterations in the hypothalamic-pituitary axis, including increased GnRH pulse frequency and dysregulated gonadotropin secretion. In this context, the study by Abbara et al. highlights that the endocrine response to GnRH agonist triggering, particularly the FSH response, may be attenuated in women with PCOS compared to healthy controls, indicating potential variability in clinical outcomes [13]. While the use of GnRH agonist triggers offers clear clinical benefits for PCOS patients undergoing IVF, accurately predicting suboptimal oocyte yield remains a significant challenge. Identifying reliable pre-trigger predictors could enhance patient selection and improve treatment outcomes. Nevertheless, the existing literature presents inconsistent findings, and studies focusing exclusively on PCOS cohorts are notably scarce.

In the current study, we aimed to present OPF as the primary outcome in patients with PCOS, when a single bolus of GnRH agonist was used for trigger during IVF treatment. As secondary outcome parameters, we determined factors that might predict the effectiveness of the GnRH agonist trigger method with respect to the OPF.

## Materials and methods

### Study population

This was a prospective study in IVF patients with polycystic ovary syndrome (PCOS) co-treated with a GnRH antagonist in which a GnRH agonist trigger was used for final oocyte maturation. The study was performed between 30/04/2021 and 30/12/2022 at Bahçeci Fulya IVF Center. PCOS was diagnosed according to International Evidence-based Guideline (Rotterdam Criteria), which requires at least two of the following: 1) oligo/anovulation with the presence of menstrual irregularity; 2) clinical evidence of hyperandrogenism (as documented by presence of a score of 6 or higher with regard to m-FG scoring); or 3) polycystic ovarian morphology (>20 antral follicle and/or >10 mm3 ovarian volume as estimated with 0.5 × length × width × depth, at least in one ovary) [14].

Exclusion criteria were as follows: female age > 35 years, body mass index (BMI) >35 kg/m2, utilization of primed ovarian stimulation protocol, preference of human chorionic gonadotropin with or without GnRH agonist for triggering ovulation and a history of ovarian surgery. *The flowchart of patient inclusion and exclusion is presented in* Fig 1.

### Ethical approval

The study was approved by the Uskudar University Ethical Committee (protocol number 61351342/Nisan 2021−29) and written informed consent was obtained from all individual participants included in the study.

### Treatment protocol

Since the study population had irregular menstrual cycles, progestin was used to induce withdrawal bleeding. Medroxyprogesterone-acetate (MPA) was used for six days with a daily dose of 10 mg (Tarlusal, Deva, Istanbul, Turkiye). Ovarian stimulation (OS) was achieved using recombinant follicle-stimulating hormone (Gonal F, Merck, Germany), with doses individualized according to BMI: 150 IU for patients with BMI < 25 kg/m2; 225 IU for patients with 25–30 kg/m2; and 300 IU for patients with BMI 30–35 kg/m2. GnRH antagonist (Cetrotide, Merck, Germany) co-treatment started on the sixth day of stimulation. Final oocyte maturation was performed using 0.2 mg of triptroline (Gonapeptyl, Ferring, Sweden) when three leading follicles exceeding 17 mm were visualized by transvaginal ultrasound. Oocyte retrieval was performed 35 hours after triggering. During the oocyte retrieval process, the number of follicles aspirated was counted and recorded.

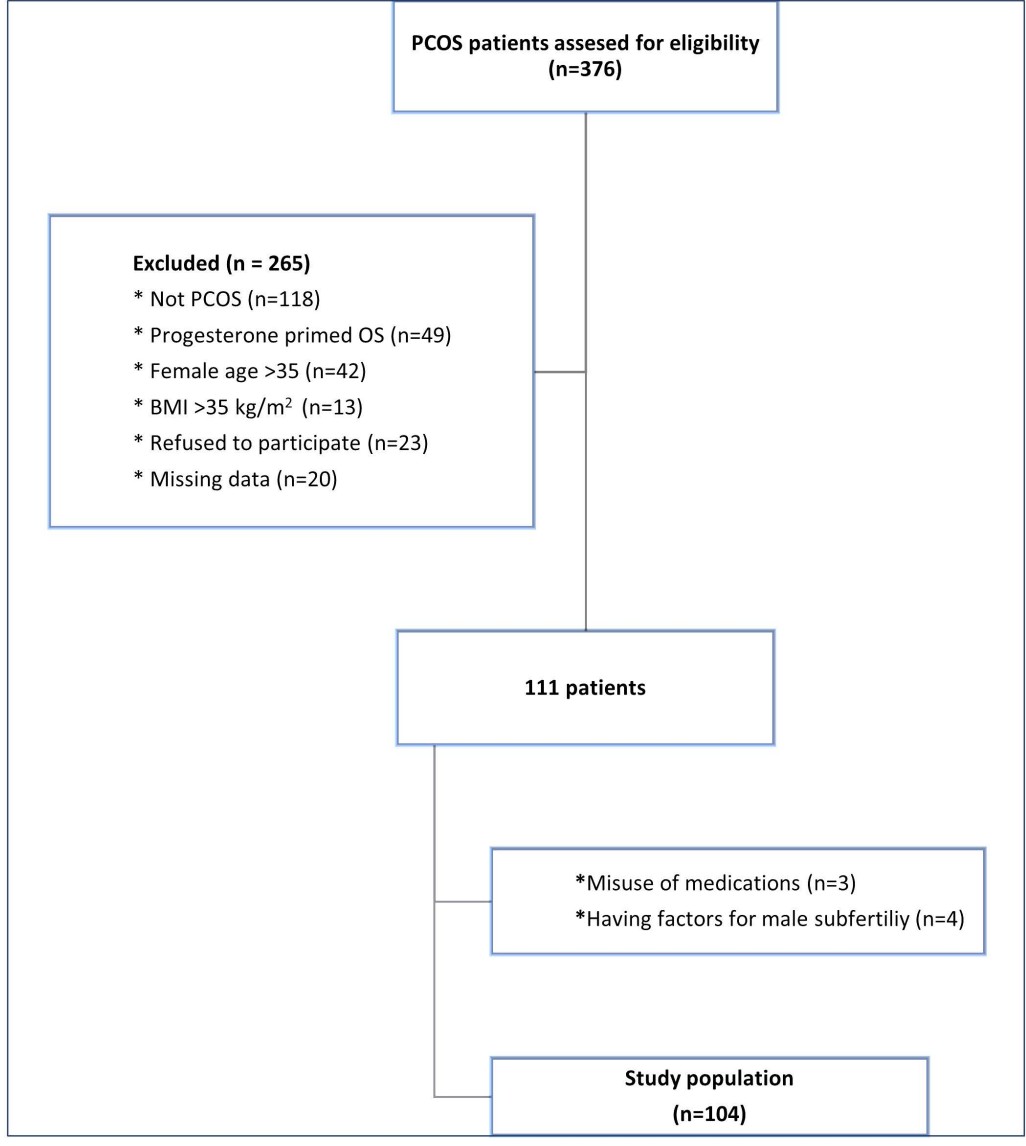

**Fig 1. Flow-chart of study.**

## Ovarian response assessment

The primary outcome parameter, the OPF was calculated by the number of oocyte cumulus complexes retrieved/number of follicles aspirated during oocyte retrieval x 100. During the oocyte retrieval collection process, each aspirated follicle (> 12 mm) was counted real-time by a single physician (FKB). To explore variations in ovarian response, patients were stratified into three groups based on the OPF values. Group 1 (Q1) represented the 0–25th percentile, Group 2 (Q2) represented the 26–75th percentile and Group 3 (Q3) represented the 76–100th percentile.

For comparison, follicular output rate (FORT, number of pre-ovulatory follicles ≥15 mm/antral follicle count x 100; 2) and follicle to oocyte index (FOI, the number of oocytes cumulus complexes retrieved/AFC x 100) were also calculated and used for comparisons.

## Assessment of patient characteristics

Anti-müllerian hormone (AMH) was measured and antral follicle counts (AFC) was recorded during the initial screening in the early follicular phase of the menstrual cycle. AMH measurement was performed using an automated system using Elecsys AMH kits (Cobas).

AFC was calculated by transvaginal ultrasonography of both ovaries by the same clinician (FKB), using a 7-MHz multi-frequency transvaginal probe (VolusonTM S6 system, GE Healthcare). Follicles measuring 2–9 mm were included in the AFC.

## Statistical analysis

Distribution characteristics of variables were visually assessed using histograms, box plots and Q–Q plots, and analyzed using Kolmogorov–Smirnov and Shapiro-Wilk tests. Continuous variables were expressed as mean ± SD or median and interquartile range, as might be needed. Numerical comparisons were made using differences in continuous variables between groups were compared using analysis of variance or the Kruskal-Wallis test, depending on data normality. Chi-squared test was used to compare the categorical variables. Pearson correlation coefficient analysis was performed to assess the relationships between OPF and continuous variables. Logistic regression analysis was performed to delineate the covariates affecting OPF. SPSS was used for statistical analysis.

## Results

A total of 104 PCOS patients were recruited according to the inclusion and exclusion criteria (Fig 1). All patients had prospectively undergone ovarian stimulation, GnRH antagonist co-treatment and trigger with a GnRH agonist.

Baseline characteristics, endocrine profiles of the participants and ovarian stimulation parameters are shown in Table 1. The mean age of the study population was 28.6 ± 3.9 years, and the mean BMI was 26.3 ± 5.4 kg/m2. On the trigger-day, the mean estradiol (E2), progesterone (P4), and luteinizing hormone (LH) levels were 5682.7 ± 3196.9 pg/ml, 1.2 ± 0.7 ng/ml and 3 ± 2.5 IU/L, respectively. The mean OPF was 70.2 ± 19.1%. The mean FORT and FOI were 49.8 ± 19.8%, and 66.2 ± 23.9%, respectively.

Patients were classified into three groups depending on their OPF rate: Q1 (0–25th percentile): 0–55%; Q2 (26–75th percentile): 56–87%, Q3 (76–100th percentile): 88–100%. The patient and cycle characteristics of these three groups are presented in Table 2. As expected, the numbers of oocytes retrieved, and mature oocytes were significantly lower in the Q1 group compared to Q2 and Q3 and were also significantly lower in Q2 compared to Q3 (Table 2). Whereas the mean female age and AMH were similar between the three groups, BMI and AFC were significantly higher in the Q1 group compared to the Q2 and Q3 groups (Table 2). Similarly, FOI and E2 per follicle significantly increased from Q1 through Q3. Although the difference was not significant, there was a trend for a decrease in trigger-day LH levels from Q1 through Q3.

As shown in Table 3, the AFC was negatively correlated with the OPF (R: −0.231, p = 0.018), whereas the FOI positively correlated with the OPF (R: 0.647, p = 0.001). When female age, AFC, BMI, duration of stimulation, total dose of gonadotropins, estradiol and LH on the day of trigger were tested, using multivariate linear regression analysis, only AFC remained as an independent predictor of the OPF (RR: −0.48, 95% CI: −0.77 to −0.19, p = 0.001) (Table 4). However, when its predictive efficacy to recognize low OPF was tested with area curve analysis, the validity was poor (AUC = 0.561).

## Discussion

This study aimed to identify factors affecting the efficacy of GnRHa trigger in patients with PCOS. Although patients with a higher AFC was at a higher risk of a suboptimal oocyte yield per aspirated follicle when triggered with an GnRH agonist, the use of AFC as a predictor appeared to be limited. Moreover, we were not able to identify any demographic, clinical, or hormonal markers as predictors for the oocyte OPF rate.

**Table 1. Baseline characteristics of the study population (n = 104).**

| Characteristic | Mean ± SD |
|---|---|
| Age (years) | 28.6 ± 3.9 |
| BMI (kg/m2) | 26.3 ± 5.4 |
| AFC | 35.7 ± 14.4 |
| AMH (ng/ml) | 4.9 ± 2.9 |
| Duration of stimulation (days) | 11.1 ± 1.5 |
| Total gonadotropin dosage (IU) | 2321.32 ± 746.46 |
| Trigger day E2 (pg/ml) | 5682.7 ± 3196.9 |
| Trigger day P4 (ng/ml) | 1.2 ± 0.7 |
| Trigger day LH (IU/L) | 3 ± 2.5 |
| No of follicles aspirated | 31.4 ± 10.2 |
| No of oocytes retrieved | 21.7 ± 8.9 |
| No of M2 oocytes | 17.25 ± 8.47 |
| Maturation rate | 79.75 ± 15.53 |
| FORT (%) | 49.4 ± 19.8 |
| FOI (%) | 66.2 ± 23.9 |
| OPF (%) | 70.2 ± 19.1 |
| E2 per follicle | 181.5 ± 73.7 |
| E2 per oocyte | 272.5 ± 115.7 |

Values are given as mean ± SD, unless stated otherwise. AFC: antral follicle count, AMH: anti-Müllerian hormone, BMI: body mass index, E2: estradiol, FOI: follicle to oocyte index, FORT: follicular output rate, LH: luteinizing hormone, OPF: oocyte per follicle, P4: progesterone.

Pituitary suppression with a GnRH antagonist has become the gold standard in IVF as it can be followed by a GnRH agonist trigger which practically eliminates the risk of ovarian hyperstimulation syndrome [6]. Although for hCG trigger, the body mass index could impact the hCG dose required to sustain high oocyte maturity rates [15], the pre-trigger criteria for predicting a suboptimal response to GnRH agonist trigger have not been yet clearly defined – and specifically not in patients with PCOS. Initially, the clinical outcome parameters associated with a suboptimal response to GnRHa trigger have been reported roughly as empty follicle syndrome or low oocyte yield, which is calculated as the number of follicles >10 mm divided by the number of oocytes retrieved [7–10]. However, a possible technical problem during oocyte retrieval and the policy of aspirating follicles with regard to their diameter might impact those outcome parameters and hence might lead to an inaccurate assessment of the trigger efficacy. In addition, there might be a trend of greater measurement errors in larger follicles during monitoring of follicular growth [11] and a steady decrease in the reliability of follicular measurements as the number of follicles increases [12]. On the basis of this, we decided to use the OPF to evaluate the effectiveness of a GnRH agonist trigger as well as analyzing factors that could influence the OPF rate.

Since an intact hypothalamic-pituitary axis is essential for efficacious triggering with a GnRHa, each patient's clinical history should be considered before planning the choice of medication. Previous studies assessed the efficacy of GnRH agonist trigger in unselected groups of infertile women undergoing IVF treatment or in oocyte donation cycles. Nevertheless, given the fact that the etiology of PCOS is thought to involve altered GnRH pulsatility, as well as ovarian and adrenal steroid dysgenesis [16,17], the response to a GnRHa trigger, particularly in patients with PCOS need to be explored. In this line, the current study is the first to present data from patients with PCOS, only, triggered with a bolus of GnRHa. To standardize the study population and suppress the high basal LH levels [18], we preferred to pretreat with MPA for six days and then initiated the OS on the second or third day of the withdrawal bleeding.

**Table 2. Comparison of patient and cycle characteristics according to OPF quartiles.**

|  | Q1 (0–25th percentile): | Q2 (26–75th percentile): | Q3 (76–100th percentile): | P value |
|---|---|---|---|---|
| Corresponding OPF (%) | 0 - 55 | 56–78 | 88 - 100 | – |
| Age (years) | 27.9±3.5 | 28.5±3.9 | 29.4±4.4 | 0.48 |
| BMI (kg/m2) | 29±7.1 | 25.1±4.3 | 25.8±4.5 | 0.029 |
| AFC | 39.8±19.2 | 35.8±13.5 | 31.4±9.1 | 0.026 |
| Duration of stimulation (days) | 11.3±1.8 | 11.2±1.3 | 10.9±1.7 | 0.43 |
| Total gonadotropin dosage (IU) | 2259±775 | 2416±833 | 2192±486 | 0.53 |
| Trigger day E2 level (pg/ml) | 4724±2056 | 5941±2700 | 6123±4659. | 0.11 |
| Trigger day P4 level (ng/ml) | 1.1±0.4 | 1.2±0.7 | 1.3±1.0 | 0.89 |
| Trigger day LH level (mIU/mL) | 3.6±2.8 | 2.9±2.5 | 2.6±2.1 | 0.34 |
| No of follicles aspirated | 32.3±8.0 | 32.5±10.6 | 29.1±11.2 | 0.054 |
| No of oocytes retrieved | 13.9±4.0 | 23.5±7.7 | 26±10 | <0.001 |
| No of MII | 11.9±3.8 | 18.3±7.7 | 20.6±10.7 | <0.001 |
| Maturation rate | 84.9±8.6 | 78.1±18.3 | 77.9±14.3 | 0.29 |
| FORT (%) | 49.1±18.6 | 50.1±20 | 50±20.1 | 0.16 |
| FOI (%) | 42.8±22.5 | 70.1±19.9 | 81.9±13.2 | <0.0001 |
| OPF (%) | 43.4±8.7 | 72.3±7.4 | 92.9±4.8 | <0.0001 |
| E2 per follicle | 145±49 | 188±80 | 204±69 | 0.003 |
| E2 per oocyte | 344±129 | 261.83±109.18 | 221.62±75.93 | 0.003 |

Values are given as mean±SD, unless stated otherwise. BMI: body mass index, AFC: antral follicle count, AMH: anti-Müllerian hormone, BMI: body mass index, E2: estradiol, FOI: follicle to oocyte index, FORT: follicular output rate, LH: luteinizing hormone, OPF: oocyte per follicle, P4: progesterone.

**Table 3. Pearson correlation analysis of variables with the ratio of oocyte per aspirated follicle.**

| Variable | Oocyte per aspirated follicle | |
|---|---|---|
|  | R | p |
| Female age (years) | 0.095 | 0.335 |
| BMI (kg/m2) | −0.16 | 0.105 |
| AMH (ng/ml) | −0.167 | 0.102 |
| AFC | −0.231 | 0.018 |
| Duration of stimulation (days) | −0.058 | 0.552 |
| Total gonadotropin dosage (IU) | 0.002 | 0.982 |
| Trigger day E2 level (pg/ml) | 0.131 | 0.185 |
| Trigger day P4 level (ng/ml) | 0.052 | 0.602 |
| Trigger day LH level (IU/L) | −0.167 | 0.101 |

AFC: antral follicle count, AMH: anti-Müllerian hormone, BMI: body mass index, E2: estradiol, LH: luteinizing hormone, OPF: oocyte per follicle, P4: progesterone.

Low baseline LH levels (<15 mIU/ml), 12 hours after GnRHa trigger have been suggested to be an independent predictor of a suboptimal oocyte yield, following the use of a GnRH agonist trigger [10]. However, the authors defined the oocyte yield as the ratio between the total number of oocytes retrieved and the number of follicles >10 mm on the day of the trigger, and this definition might be influenced by technical difficulties during oocyte retrieval, as mentioned above.

Other than the initial visits, LH levels might also fluctuate during OS, depending on the medication used for pituitary suppression [19,20]. In this context, low LH levels at the start of OS may be an indirect indicator of hypothalamic

**Table 4. Multivariate linear regression analysis of factors affecting oocyte per aspirated follicle.**

| Variables | B coefficient | P value |
| --- | --- | --- |
| Age | 0.27 (−0.75 to 1.299) | 0.596 |
| AFC | −0.48 (−0.77 to −0.19) | 0.001 |
| Duration of stimulation | 0.42 (−2.35 to 3.19) | 0.764 |
| BMI | −0.59 (−1.32 to 0.15) | 0.118 |
| Trigger day E2 | 0.002 (0.001 to 0.004) | 0.07 |
| Trigger day P4 | −1.01 (−6.78 to 4.76) | 0.730 |
| Trigger day LH | −1.71 (−3.20 to 0.060) | 0.059 |

AFC: antral follicle count, BMI: body mass index, E2: estradiol, LH: luteinizing hormone, P4: progesterone.

suppression and may lead to temporary hypothalamic-pituitary dysfunction. The main causes of hypothalamic suppression are long-term use of oral contraceptives and low BMI [9]. In this line, Meyer et al. performed a retrospective analysis of risk factors for suboptimal response, which was defined by LH level of <15 mIU/mL, 12 hours after the GnRH agonist trigger. Although the percentage of suboptimal responses was higher in cases with lower LH levels at the start of OS and on the trigger day, multivariate regression analysis showed that long-term oral contraceptive use was the only independent risk factor [9]. As low LH levels on the trigger day might result from long-term hypothalamic-pituitary axis suppression, these cases might be associated with a mild form of a non-intact hypothalamic-pituitary axis. Our study evaluated a homogenous PCOS study population, and the pretreatment strategy was standardized by using MPA, as mentioned previously. In contrast to previous studies, the lack of any finding of a LH threshold on the day of triggering might be related to the primary outcome of our study or suppression of basal LH with a pretreatment of MPA for six days.

The association between AFC and the risk of suboptimal oocyte yield per aspirated follicle warrants further investigation to elucidate the underlying mechanisms. Notably, Genro et al. demonstrated an inverse relationship between anti-Müllerian hormone (AMH) levels and the follicular output rate (FORT), attributing this to the inhibitory role of AMH on certain putative follicle-stimulating hormone (FSH) actions in ovarian follicles [21]. These findings have been corroborated by additional human and animal studies, which have also reported the inhibitory effects of AMH on natural ovulation [22]. Although AMH levels were not measured in the present patient cohort—owing to the routine preference in our institution for antral follicle count over AMH assessment prior to stimulation—extensive evidence underscores the strong correlation between these two ovarian reserve parameters [23]. However, a high AFC in itself has been also reported to be associated with lower (10th percentile) oocyte recuperation rate within a cohort of 1747 women treated with GnRH antagonist when 0.2 mg of triptorelin had been preferred for triggering [24]. An unexpectedly poorer oocyte retrieval performance in patients with PCOS may also be attributed to a distinct endogenous FSH and LH secretion pattern following GnRHa triggering, compared to otherwise healthy individuals and those with hypothalamic amenorrhea [13]. Although triptorelin administration induced a comparable peak in serum LH across all groups (mean peak LH: healthy, 52.3 IU/L; PCOS, 46.2 IU/L; hypothalamic amenorrhea, 41.3 IU/L), the area under the curve (AUC) for the change in serum FSH was significantly attenuated in women with PCOS (median AUC for FSH change: PCOS, 127.2; healthy, 253.8; hypothalamic amenorrhea, 326.7 IU.h/L; p = 0.0005) [13]. These findings raise the hypothesis that supplementing the attenuated endogenous FSH surge with exogenous FSH at the time of GnRHa-induced ovulation might enhance oocyte retrieval rates and oocyte maturity. Consequently, elevated AFC and high AMH levels in PCOS patients may further impair oocyte yield by inhibiting follicular growth and ovulation, and by disrupting the FSH surge dynamics during ovulation induction.

With regard to BMI of the patients, several studies have consistently demonstrated a statistically significant negative correlation with the outcome of IVF success parameters [25]. In a recent study including 981 cycles, the risk factors for a suboptimal response to GnRH agonist trigger was reported to be: a high BMI, lower LH levels through the OS, prolonged

stimulation and the need for high exogenous gonadotropin dosing [26]. Similarly, in our data, a comparison of demographic characteristics in high, moderate and low OPF groups revealed that BMI was negatively correlated with OPF. Although BMI was not found to be a statistically significant factor in multivariate linear regression analysis, it should be tested in a larger sample size.

The E2 level on the day of triggering serves as an indicator of the follicular response to exogenous gonadotropins, with evidence suggesting a linear relationship between the E2 response and the efficacy of GnRH agonist triggering. This aligns with the observation that the E2 level per follicle significantly increased from the first to the third quartile of ovarian performance. This may imply that luteinized follicles, which produce a robust E2 response following GnRH agonist trigger, are also associated with a higher ovarian performance.

Evaluating post-trigger hormonal responses—specifically serum LH and FSH levels—represents a critical step toward understanding the physiological mechanisms underlying the efficacy of GnRH agonists in PCOS patients. Considering that corrective options are limited once a suboptimal hormonal response is detected after the trigger, it becomes increasingly important to focus on the factors within our control prior to triggering. In our clinical practice, therefore, comprehensive evaluation of pre-trigger parameters—including AFC, baseline LH, BMI, and patient history—is emphasized to anticipate the likelihood of a suboptimal response.

Random start protocols may be considered a feasible approach in PCOS patients due to their frequently irregular cycles. However, in this study, we preferred to initiate ovarian stimulation following MPA-induced withdrawal bleeding to achieve standardization across participants. Beyond facilitating uniform cycle initiation, the use of MPA may also contribute to the suppression of elevated basal LH levels, which are commonly observed in PCOS and could otherwise confound the interpretation of trigger-day endocrine parameters.

The main limitation of the present study is the relatively small sample size, however the inclusion of only PCOS patients in a prospective setting and the utilization of a more valid outcome parameter are novel factors compared to the existing literature. A secondary drawback is the lack of a post-trigger LH level measurement.

In conclusion, although GnRHa for triggering final oocyte maturation is safe for PCOS patients co-treated with a GnRH antagonist, the present study did not detect any powerful demographic, clinical nor endocrinological tool to predict the efficacy of the trigger. The risk of a suboptimal response after a GnRH agonist trigger still remains a possibility in PCOS patients and the AFC yields only a poor performance for the prediction of the OPF rate.

## Author contributions

**Conceptualization:** Fazilet Kubra Boynukalin, Zalihe Yarkıner, Gürkan Bozdag.

**Data curation:** Fazilet Kubra Boynukalin, Meral Gultomruk, Özkan Özdamar.

**Formal analysis:** Zalihe Yarkıner, Meral Gultomruk, Gürkan Bozdag.

**Methodology:** Fazilet Kubra Boynukalin, Özkan Özdamar, Gürkan Bozdag.

**Writing – original draft:** Fazilet Kubra Boynukalin, Peter Humaidan, Mustafa Bahçeci, Gürkan Bozdag.

**Writing – review & editing:** Fazilet Kubra Boynukalin, Peter Humaidan, Mustafa Bahçeci, Gürkan Bozdag.

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
