## [Decision Letter · Decision Letter 0]

15 Aug 2025

Dear Dr. Boynukalin,

Thank you for submitting your manuscript to PLOS ONE. After careful consideration, we feel that it has merit but does not fully meet PLOS ONE’s publication criteria as it currently stands. Therefore, we invite you to submit a revised version of the manuscript that addresses the points raised during the review process.

2- Manuscript content: Rationale is missing which should be part of the Introduction

We look forward to receiving your revised manuscript.

Kind regards,

Wan-Xi Yang, Ph.D.

Academic Editor

PLOS ONE

**Journal Requirements:**

1. When submitting your revision, we need you to address these additional requirements. Please ensure that your manuscript meets PLOS ONE's style requirements, including those for file naming. The PLOS ONE style templates can be found at https://journals.plos.org/plosone/s/file?id=wjVg/PLOSOne_formatting_sample_main_body.pdf and https://journals.plos.org/plosone/s/file?id=ba62/PLOSOne_formatting_sample_title_authors_affiliations.pdf 2. In the online submission form, you indicated that your data will be submitted to a repository upon acceptance.  We strongly recommend all authors deposit their data before acceptance, as the process can be lengthy and hold up publication timelines. Please note that, though access restrictions are acceptable now, your entire minimal  dataset will need to be made freely accessible if your manuscript is accepted for publication. This policy applies to all data except where public deposition would breach compliance with the protocol approved by your research ethics board. If you are unable to adhere to our open data policy, please kindly revise your statement to explain your reasoning and we will seek the editor's input on an exemption. 3. When completing the data availability statement of the submission form, you indicated that you will make your data available on acceptance. We strongly recommend all authors decide on a data sharing plan before acceptance, as the process can be lengthy and hold up publication timelines. Please note that, though access restrictions are acceptable now, your entire data will need to be made freely accessible if your manuscript is accepted for publication. This policy applies to all data except where public deposition would breach compliance with the protocol approved by your research ethics board. If you are unable to adhere to our open data policy, please kindly revise your statement to explain your reasoning and we will seek the editor's input on an exemption. Please be assured that, once you have provided your new statement, the assessment of your exemption will not hold up the peer review process. 4. If the reviewer comments include a recommendation to cite specific previously published works, please review and evaluate these publications to determine whether they are relevant and should be cited. There is no requirement to cite these works unless the editor has indicated otherwise. 

Reviewers' comments:

**Comments to the Author**

1. Is the manuscript technically sound, and do the data support the conclusions?

Reviewer #1: Yes

Reviewer #2: Yes

2. Has the statistical analysis been performed appropriately and rigorously?

Reviewer #1: Yes

Reviewer #2: Yes

3. Have the authors made all data underlying the findings in their manuscript fully available?

Reviewer #1: Yes

Reviewer #2: Yes

4. Is the manuscript presented in an intelligible fashion and written in standard English?

Reviewer #1: Yes

Reviewer #2: Yes

**Reviewer #1:**  A much needed topic. A well written Manuscript supported with evidence based data.

1- Abstract: The short Background is missing.

2- Manuscript content: Rationale is missing which should be part of the Introduction

3- Discussion: Recommendations need to be added after the limitations of the study.

**Reviewer #2: ** Determining predictors for poor responses in PCOS is certainly a challenge and I applaud the authors for their attempt, even if it was not the outcome they may have hoped for. I would suggest another limitation is the use of MPA prior to stim start instead of utilizing laboratory values to ensure patients were at the same phase of cycle (ie early follicular) vs. a "random start."

**Do you want your identity to be public for this peer review?** For information about this choice, including consent withdrawal, please see our Privacy Policy

Reviewer #1: No

Reviewer #2: **Yes: ** Sara J. Mucowski, M.D.

---

## [Author Response · Author response to Decision Letter 1]

1 Sep 2025

Dear Editor,

We sincerely thank the Academic Editor and the reviewers for their thoughtful and constructive comments. We have carefully considered each point and revised the manuscript accordingly. Below, we provide a detailed response to every comment, indicating how and where changes were made in the revised manuscript. We hope that the revisions and clarifications adequately address all concerns and improve the quality and clarity of our work.

Response to Reviewers

Reviewer #1:

Comment 1: "Abstract: The short Background is missing."

Author Response:

We thank the reviewer for this insightful comment. In accordance with the suggestion, we have revised the abstract to include a concise background section. The revised version now briefly explains the clinical relevance of GnRH agonist triggers in PCOS patients and the rationale for investigating predictors of oocyte yield. This addition helps contextualize the study aim for readers from the outset.

Revised Abstract Opening:

“Background: Polycystic ovary syndrome (PCOS) is associated with altered hypothalamic-pituitary-ovarian function, which may affect the success of GnRH agonist triggers used during IVF. Identifying reliable predictors of oocyte yield in these patients remains a clinical challenge.”

We hope this revision meets the reviewer’s expectations and improves the clarity and completeness of the abstract.

Comment 2: "Manuscript content: Rationale is missing which should be part of the Introduction."

Author Response:

We appreciate the reviewer’s comment regarding the importance of clearly presenting the rationale in the Introduction section. To address this, we have added a concise explanation highlighting the current uncertainty in identifying reliable predictors of oocyte yield in PCOS patients undergoing GnRH agonist trigger. This paragraph now establishes the unmet clinical need and positions our study as a focused attempt to address this gap.

Added Text in the Introduction (last paragraph):

While the use of GnRH agonist triggers offers clear clinical benefits for PCOS patients undergoing IVF, accurately predicting suboptimal oocyte yield remains a significant challenge. Identifying reliable pre-trigger predictors could enhance patient selection and improve treatment outcomes. Nevertheless, the existing literature presents inconsistent findings, and studies focusing exclusively on PCOS cohorts are notably scarce.

Determining predictors for poor responses in PCOS is certainly a challenge and I applaud the authors for their attempt, even if it was not the outcome they may have hoped for. I would suggest another limitation is the use of MPA prior to stim start instead of utilizing laboratory values to ensure patients were at the same phase of cycle (ie early follicular) vs. a "random start."

We appreciate the reviewer’s insightful comment regarding the need to further elaborate on the clinical implications of post-trigger hormonal assessment and the potential limitations in intervention once the trigger has been administered.

In response, we have added the following paragraph to the Discussion section to clarify our clinical rationale and emphasize the importance of pre-trigger evaluation:

“Considering that corrective options are limited once a suboptimal hormonal response is detected after the trigger, it becomes increasingly important to focus on the factors within our control prior to triggering. In our clinical practice, therefore, comprehensive evaluation of pre-trigger parameters—including AFC, baseline LH, BMI, and patient history—is emphasized to anticipate the likelihood of a suboptimal response. This proactive strategy is particularly relevant for PCOS patients, in whom altered gonadotropin dynamics may not always translate into predictable post-trigger outcomes.”

We believe this addition strengthens the clinical relevance of our findings and addresses the reviewer’s concern.

Response to Reviewer 2:

We sincerely thank the reviewer for the valuable observation regarding the use of MPA prior to stimulation and the potential implications for cycle phase standardization.

We agree that initiating stimulation based on laboratory-confirmed early follicular phase could offer tighter control over the baseline hormonal milieu. However, in patients with PCOS—who often experience oligo/anovulation—this approach may not always be practical. In our study, we chose to administer MPA to induce withdrawal bleeding, thereby ensuring a standardized and timely initiation of ovarian stimulation across participants.

In addition to facilitating synchronization, the use of MPA may have also contributed to the suppression of elevated basal LH levels—a common feature in PCOS—which could otherwise interfere with the interpretation of pre-trigger hormonal profiles. This rationale has now been clarified and acknowledged as both a methodological choice and a potential limitation in the revised Discussion section.

We appreciate the reviewer’s thoughtful input, which helped improve the clarity of our design considerations.

---

## [Editor Report · Decision Letter 1]

4 Sep 2025

Limitations of hormonal and clinical markers in predicting GnRH agonist trigger success in polycystic ovary syndrome: A critical reappraisal

PONE-D-25-23181R1

Dear Dr. Boynukalin,

We’re pleased to inform you that your manuscript has been judged scientifically suitable for publication and will be formally accepted for publication once it meets all outstanding technical requirements.

Kind regards,

Wan-Xi Yang, Ph.D.

Academic Editor

PLOS ONE
---

## [Editor Report · Acceptance letter]

PONE-D-25-23181R1

PLOS ONE

Dear Dr. Boynukalin,

I'm pleased to inform you that your manuscript has been deemed suitable for publication in PLOS ONE. Congratulations! Your manuscript is now being handed over to our production team.

Kind regards,

on behalf of

Dr. Wan-Xi Yang

Academic Editor

PLOS ONE